# AN EMPIRICAL STUDY OF POLICY INTERPOLATION VIA DIFFUSION MODELS

Yuqing Xie[1], Chao Yu[1,2], Ya Zhang[3,4], Yu Wang[1]

[1]Tsinghua University, [2]Beijing Zhongguancun Academy,
[3]Shanghai Jiao Tong University, [4]Shanghai Artificial Intelligence Laboratory,

## ABSTRACT

Diffusion-based policies have shown great potential in multi-task settings, as they can solve new tasks without additional training through inference-time steering. In this paper, we explore the inference-time composition of diffusion-based policies using various interpolation methods. Our results show that, while existing methods merely switch between predefined action modes, our proposed approach can generate entirely new action patterns by leveraging existing policies, all without the need for further training or tuning.

## 1 INTRODUCTION

In real-world applications, robots often need to solve new tasks that are out of training task distribution. With behavioral cloning, we can distill multiple task-specific policies into a single model, which not only enables the robot to solve a variety of tasks with one model, but also provides certain cross-task transfer capability to tackle new tasks. (Reuss et al., 2023; Liang et al., 2024; Ma et al., 2024)

Diffusion models are generative models that have been widely applied in image generation for its high-quality generation and training stability (Ho et al., 2020; Kingma et al., 2021; Du & Kaelbling, 2024). When applied to robot policies, diffusion models are able to represent multimodal action distributions and handle high-dimensional action spaces (Chi et al., 2023), which significantly improves policy performance. Additionally, diffusion-based policies facilitate task composition during the inference phase, either through reward signals (Janner et al., 2022; Wang et al., 2022) or action constraints (Mishra et al., 2023).

Among these methods, Decision Diffuser (DD) (Ajay et al., 2022) achieves a preliminary form of skill composition. DD diffuses over the state sequence, and use an inverse dynamics model to predict actions from neighboring states. When trained to imitate various gaits of a quadruped robot, DD can generate a new gait that combines the characteristics of all the trained gaits with inference-time Classifier-Free Guidance (CFG) (Ho & Salimans, 2022).

However, DD only switches between different behavior modes without efficiently merging them. To facilitate policy interpolation, we design and compare different inference-time interpolation methods for diffusion-based policies. We also incorporates CFG++ technique (Chung et al., 2024), which shows improvement in image generation tasks, into the our sampling procedure. The results show that CFG, with proper diffusion modeling and sampler, can blend existing policy modes into new ones without further training or tuning, and thus achieve skill composition via policy interpolation.

## 2 METHOD

Diffusion process composes of two sub-processes. The forward process gradually adds Gaussian noise to the initial data point $x_0$, $x_t = \alpha_t x_0 + \sigma_t \epsilon$, where $\alpha_t$ and $\epsilon$ are predefined noise schedule. The reverse process iteratively denoises current sample $x_t$ to obtain $x_0$. For example, using probability flow ODE (PF-ODE) (Song et al., 2020b), we obtain:

$$\hat{x}_t = (x_t - \sqrt{1 - \bar{\alpha}_t})\epsilon_t / \sqrt{\bar{\alpha}_t} \tag{1}$$

$$x_{t-1} = \sqrt{\bar{\alpha}_{t-1}}\hat{x}_t + \sqrt{1 - \bar{\alpha}_{t-1}}\epsilon_t \tag{2}$$

where $\epsilon_t = \epsilon_t(x_t, t)$ is parameterized by a neural network that conditioned on current data point $x_t$ and sampling time step $t$, and $\bar{\alpha}_t$ is determined by $\alpha_t$.

For guided sampling, we employ CFG, which substitutes $\epsilon_t(x_t, t)$ with $\hat{\epsilon}_t(x_t, t, y) = \epsilon_t(x_t, t, \emptyset) + w(\epsilon_t(x_t, t, y) - \epsilon_t(x_t, t, \emptyset))$ in equation 1 and 2, where $y$ is the condition and $w$ is guidance weight. We train the network with certain condition dropout rate to obtain the conditioned and the unconditioned $\epsilon_t$ simultaneously. CFG++ further substitutes equation 2 with $x_{t-1} = \sqrt{\bar{\alpha}_{t-1}}\hat{x}_t + \sqrt{1 - \bar{\alpha}_{t-1}}\epsilon_t(x_t, t, \emptyset)$ to achieve better sample quality and reduce mode collapse in image generation. We incorporate CFG++ sampling to our diffusion backbone.

We consider our problem as conditioned imitation learning. For clarity, in the following notations, we omit the subscript $t$ in the sampling process and use $t$ for the time step of a given trajectory. Given a dataset $D = \{\tau_k | \tau_k = (s_t, a_t, y)_{t=1}^{N_k}\}$ that consists of states $s$, actions $a$, and one-hot task labels $y$, we diffuse over trajectory segments, $x = ((s_t, a_t)|f(y))$, where $f$ is a network that encodes $y$ to latent space, and $y_1|y_2$ is the union of the one-hot labels $y1, y2$.

We will compare 4 different policy interpolation methods, together with Decision Diffuser (DD):

- Noise Model Merge (NM), where $\hat{\epsilon}_{NM} = 0.5\epsilon_t(x, f(y_1)) + 0.5\epsilon_t(x, f(y_2))$;
- Direct Label Merge (DL), where $\hat{\epsilon}_{DL} = \hat{\epsilon}_t(x, f(y_1|y_2))$;
- Latent Label Merge (LL), where $\hat{\epsilon}_{LL} = \hat{\epsilon}_t(x, 0.5f(y_1) + 0.5f(y_2))$;
- CFG Merge (CFG), where $\hat{\epsilon}_{CFG} = 0.5\hat{\epsilon}_t(x, f(y_1)) + 0.5\hat{\epsilon}_t(x, f(y_2))$.

## 3 EXPERIMENTS

We benchmark the interpolation methods in MuJoCo HalfCheetah environment. We design three base tasks, where the robot moves forward at speeds of 1, 2, and 3 m/s, and two interpolating tasks with forward speeds of 1.5 and 2.5. We collect expert datasets for each base task following D4RL pipeline as in Fu et al. (2020) and then train all models via imitation learning until convergence.

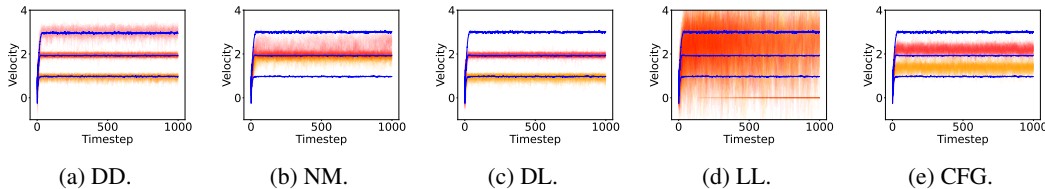

|     (a) DD.     |     (b) NM.     |     (c) DL.     |     (d) LL.     |     (e) CFG.    |

Figure 1: The performance of five different policy interpolation methods. Each curve represents a collection of 50 rollout trajectories for a given task. The x-axis denotes the time steps within the episode, while the y-axis denotes the robot's current speed. The blue lines depict the average trajectories for the base tasks (vel=1, 2, 3). The red line shows the interpolation result between vel=2 and vel=3, and the orange line shows the interpolation result between vel=1 and vel=2.

We present the results in Figure 1. The interpolated trajectories generated by DD are consistent with the base tasks trajectories. In other words, DD randomly selects one of the two base task trajectories as the output. We hypothesize that the inverse dynamics model limits the generalization capability of the diffusion model, and thus making the output actions heavily rely on the existing dataset distribution. NM produces interpolated trajectories that are similar to the base task trajectories, lacking strong guidance towards new action modes. Methods like DL, which merely modifies the condition labels, can only select trajectories within base task trajectories. Due to the lack of regulations on the latent label space, the conditions fused by LL are semantically invalid, and therefore produces invalid actions after guided sampling. In contrast, our simple model with CFG merging integrates the data gradients from both base tasks during the denoising process. As a result, it successfully finds new action modes and generates stable trajectories that lie between the base task trajectories.

We further quantitatively analyze the impact of CFG++ on the quality of interpolation. We compute the average deviation between the interpolated trajectory and the intermediate trajectory directly

merged by two base trajectories. The deviation for CFG is 0.302, while CFG++ reaches 0.197. This indicates that CFG++ improves sample quality and yields more accurate policy interpolation results.

## URM STATEMENT

The authors acknowledge that at least one key author of this work meets the URM criteria of ICLR 2025 Tiny Papers Track.

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

# A APPENDIX

## A.1 EXPERIMENT SETUP

We design three base tasks for MuJoCo HalfCheetah environment, where the robot moves forward at speeds of 1, 2, and 3 m/s, respectively. To collect dataset, we train RL policy models with SAC until convergence, and then collect 1000 RL rollout trajectories for each task. We present the average performance of the dataset in Table 1.

| Target Velocity | 1 | 2 | 3 |
|---|---|---|---|
| Average Velocity in Dataset | 0.969 | 1.914 | 2.976 |

Table 1: Dataset quality.

Our diffusion-based policy is built upon CleanDiffuser (Dong et al., 2024). We use DiT1d model (Dong et al., 2023) as the diffusion model backbone and MLP as the condition encoder. The total model size is about 4MB. During the denoising process, we employ DDIM (Song et al., 2020a) with CFG++ augmentation (Chung et al., 2024) as the solver.

## A.2 LIMITATIONS AND FUTURE WORK

In this paper, we present preliminary evidence of the potential for policy interpolation using diffusion models. However, our approach lacks certain theoretical guarantees and may not generalize reliably in all settings.

Diffusion models generate plausible interpolations within a learned latent space; the interpolations are then mapped back to the task space (i.e., the physical world) to produce the interpolated policy. However, there is no theoretical guarantee of a unique or invertible mapping between the latent space and the task space. In other words, the interpolated policy may exhibit several distinct behaviors when mapped to the task space, yet they are all valid interpolations in the latent space. For example, when the learned latent space corresponds to the robot's forward velocity in the task space, interpolating between two policies results in a new policy whose velocity is a blend of the velocities of the two base policies. However, if the latent space corresponds to the robot's joint angles, the interpolated policy will be a combination of the joint angles of the two base policies. Moreover, the mapping from latent space to task space can be highly non-linear. This means that even if we specify a 1:1 interpolation in the latent space, the resulting interpolated policy may not correspond to the average of the two base policies.

We believe that future research could explore the following directions:

- Experiment on more scenarios, generate more complex diffusion models with a wider variety of base policies, and extensively evaluate the interpolation results.

- Regulate diffusion models to establish a more explicit, interpretable, and adjustable mapping between the latent space and task space.

- Properly structure the latent embedding to better capture task-specific parameters. For instance, incorporating explicit task parameters, such as velocity, directly into the latent representation, may improve the accuracy and consistency of interpolated policies.

