# OpenReview forum: "An Empirical Study of Policy Interpolation via Diffusion Models"
_ICLR.cc/2025/Workshop/MCDC — MCDC @ ICLR 2025_

### Official Review · Reviewer_rTg6 · 2025-02-16

**Rating:** 3
**Confidence:** 4
**Fit:** 3

**Summary:**

This paper is an empirical investigation into how generative diffusion policies can be used for policy interpolation. A variety of interpolation strategies are evaluated in a MuJoCo HalfCheetah environment; in the offline dataset the actor moves at a fixed speed (velocity 1, 2, 3), and the goal is to guide the conditional diffusion model to run at alternative speeds (1.5, 2.5).

**Reason For Giving A Higher Score:**

If the issue with problem formulation did not exist, all other issues are minor, and this would be a good paper to include in the workshop.

**Reason For Giving A Lower Score:**

The most important weakness of the paper is the problem formulation. I do not think the problem can be solved as it stands, and the current results will not generalize broadly (I have discussed why). This makes it challenging to find the takeaway message of the submission.

**Strengths And Weaknesses:**

---

### Strengths

I like the problem motivation, and found the introduction and findings to be concise and well-explained. Except for a few issues with formulations in §2 and figures in §3, I liked the presentation.

There did not seem to be any major errors or inconsistencies in the text.

---
### Weaknesses

Unfortunately, I think there is a significant flaw in the problem formulation that limits what we can learn from this work. I will try my best to articulate it here.

***TLDR:*** There is a fundamental reason that makes this problem impossible to solve, and strongly suggests that the empirical observations will not generalize. The only way to correct this is to rethink the problem formulation.

Suppose the trajectory distribution for velocity $v$ (task $y_v$) are identified with some one-hot encoded variable $z_v \in \\{0, 1\\}$. Based on §2, we condition the diffusion model with $f(z_1, z_2)$ (I'll ignore velocity 3 for simplicity). Note that there is no information in $z_v$ about $v$, the actual velocity, and what $f(z_1, z_2)$ learns is open-ended.

Now, the goal here was to see if you can interpolate $f(1, 0)$ (task $y_1$ with velocity 1) and $f(0, 1)$ (task $y_2$ with velocity 2) to somehow achieve speed $\frac{1+2}{2} = 1.5$. The key question to ask here, is **"if the diffusion model or $f(z_1, z_2) do not see the velocity, how should there be an interpolation that consistently guides them to 1.5?"**.

Suppose, our of sheer luck, $f(z_1, z_2)$ actually learns to encode the true speed, i.e., $f(z_1, z_2) = z_1 + z_2 \times 2$. Then, due to this linear relationship, $\frac{f(1, 0)+f(0, 1)}{2} = 1.5$ should condition the diffusion model to generate trajectories at speed 1.5, and the CFG merge function works.

But now suppose the function learns to encode the log of speed, simply because that was easier to encode, i.e., $f(z_1, z_2) = z_1 \log 1 + z_2 \times \log 2 \approx \log2 z_2$. Now, $\frac{f(1, 0)+f(0, 1)}{2} = 0.5\log2$ represents a speed of $\sqrt{2}$ to the diffusion model, and CFG merge fails. In fact, none of the merge functions will work.

What the latent encoder $f(\cdot)$ and the diffusion model will converge to is not unique, since infinitely many models will reach the same exact loss. To see why, consider any invertible function $H(\cdot): \mathcal{R}^d \rightarrow \mathcal{R}^d$. For a given latent encoder $f(z_1, z_2): \mathcal{R}^2 \rightarrow \mathcal{R}^d$, create a new encoder $\hat{f}(z_1, z_2)=H(f(z_1, z_2))$. For the corresponding diffusion model $g(\tau, c, t)$, where $c$ is the latent embedding, create a new diffusion model $\hat{g}(\tau, c, t)=g(\tau, H^{-1}(c), t)$. The pair $<\hat{f}(\cdot), \hat{g}(\cdot)>$ will achieve the same exact training loss as $<f(\cdot), g(\cdot)>$, but will behave differently on all interpolation functions, including those you consider in §2 and §3.

So essentially, for every interpolation function that mixes the tasks well in some environment for some seed, there is some training seed that adversarially breaks that merging function for the same environment. Even if some merge function works consistently well for some environment due to function regularization, it may not work on another environment.

Note that while the DD paper does something similar with merging multiple tasks, the setting there is slightly different. Each "task" is some constraint on trajectories, e.g., the actor cannot move outside of some bounded region in the 2-D state space. The merging there intends to generate trajectories that respect both bounds. From a function approximation perspective, this type of merge is more feasible to achieve consistently empirically, although it suffers from the same adversarial issue.

---

**Below are minor issues and did not affect my decision. I include them here as feedback.**

There are some issues with formalization in §2, such as Equation 2 missing a closing parenthesis, or the formula for CFG++ being incorrect.

Also, I think the visualization in Figure 1 could have been better. For example, you could plot CDFs of achieved speed across steps, where each curve is some merging approach. You would then need 2 plots at most to show everything, and this would make comparisons easier. It is also best to include the x and y labels in the figures themselves, even though you include them in the captions.

**Suggestions:**

The key issue here, in my opinion, is the problem formulation. Without specifying how you want the tasks merged, no solution exists for this problem, and this statement is not specific to diffusion models. I would suggest changing the problem formulation and revisiting this.

---

### Official Review · Reviewer_DTsb · 2025-02-17

**Rating:** 7
**Confidence:** 4
**Fit:** 4

**Summary:**

In this paper, the authors investigate how to blend or “interpolate” multiple diffusion-based robot policies at inference time to produce new, previously unseen behaviors. They focus on a multi-task setting in which each policy is trained through diffusion modeling on different, discrete tasks (e.g., varying target velocities for a HalfCheetah robot), but they want to combine these tasks without additional training or fine-tuning.

**Reason For Giving A Higher Score:**

I think the method is novel and can scale up to more practical robotics applications to allow for efficient composition and merging of existing base policies.

**Reason For Giving A Lower Score:**

I think having more experiments where tasks and the number of policies are scaled, especially in  manipulation would strengthen the paper. However, given the 2 page tiny format I understand the constraint.

**Strengths And Weaknesses:**

I find the use of classifier free guidance to merge diffusion policies unique and novel to my knowledge. The results in Figure 1 show how CFG provides a consistent merging behavior whereas DD has a harder time producing out of distribution outputs. I think it would be good to scale the experiments up for future work not just with tasks but also more policies.

**Suggestions:**

typo eq 1. missing )

---

### Official Review · Reviewer_wCrU · 2025-02-27

**Rating:** 6
**Confidence:** 3
**Fit:** 3

**Summary:**

The paper investigates merging for test-time steering of diffusion models. This is done by controlling the noise distribution that's added over the input during the iterative diffusion procedure; by making this noise distribution conditioned on additional variable $y$ (e.g. a task id or label), steering is enabled. The contributions of the paper relate to how one can merge / interpolate across two different policies that are obtained via different steering values $y1,y2$. On mujoco experiments, where data from different policies are collected (half-cheetah at different speeds) the authors examine several ways to interpolate across policies. They conclude that Classifier-Free Guidance can achieve strong policy interpolation, effectively showing that the final policy operates at an interpolated speed from the two policies.

**Reason For Giving A Higher Score:**

addressed above

**Reason For Giving A Lower Score:**

adressed above

**Strengths And Weaknesses:**

**Strengths**
1. The experiment proposed is the paper is clear and shows that CFG achieves the expected goal of policy interpolation
2. For a 2-page paper, the authors did a good job at presenting it

**Weaknesses**
1. It would be much better if the authors can expand the current version. It's unclear to me what is the contribution of this paper. Is it the application and CFG / CFG++ for policy interpolation ? What is the related work for policy interpolation with diffusion models in this setting ? The authors mention Decision Diffuser (DD), however DD is not in the experimental section
2. The paper could greatly benefit from a paragraph that details how the conditioning on the task is done. (see suggestions)

**Suggestions:**

Some questions I have, hopefully addressing them will lead to a clearer paper

What is "f" here exactly (lines 69-74) ?
What does it mean to have a specific label, or label2|label1 as input?

---

### Decision · Program_Chairs · 2025-03-06

**Decision:**

Accept

**Comment:**

This paper presents an interesting contribution on merging for test-time steering of diffusion models. The program chairs have reviewed the comment from reviewer rTg6 and strongly encourage the authors to address this point in the final version of the paper.